# Apoptotic neurodegeneration in whitefly promotes the spread of TYLCV

Shifan Wang[1,2], Huijuan Guo[1,2], Feng Ge[1,2,3]*, Yucheng Sun[1,2]*

[1]State Key Laboratory of Integrated Management of Pest Insects and Rodents, Institute of Zoology, Chinese Academy of Sciences, Beijing, China; [2]CAS Center for Excellence in Biotic Interactions, University of Chinese Academy of Sciences, Beijing, China; [3]Maoming Branch, Guangdong Laboratory of Lingnan Modern Agriculture, Maoming, China

**Abstract** The mechanism by which plant viruses manipulate the behavior of insect vectors has largely been described as indirect manipulation through modifications of the host plant. However, little is known about the direct interaction of the plant virus on the nervous system of its insect vector, and the substantial behavioral effect on virus transmission. Using a system consisting of a *Tomato yellow leaf curl virus* (TYLCV) and its insect vector whitefly, we found that TYLCV caused caspase-dependent apoptotic neurodegeneration with severe vacuolar neuropathological lesions in the brain of viruliferous whitefly by inducing a putative inflammatory signaling cascade of innate immunity. The sensory defects caused by neurodegeneration removed the steady preference of whitefly for virus-infected plants, thereby enhancing the probability of the virus to enter uninfected hosts, and eventually benefit TYLCV spread among the plant community. These findings provide a neuromechanism for virus transmission to modify its associated insect vector behavior.

*For correspondence:
gef@ioz.ac.cn (FG);
sunyc@ioz.ac.cn (YS)

Competing interests: The authors declare that no competing interests exist.

## Introduction

Arboviruses contribute to a substantial portion of the global disease burden, which can be effectively transmitted by insect vectors (*Eigenbrode et al., 2018*). Among the plant-virus-vector associations, insect vectors are the only organisms that can freely disperse and closely interact with both host plants and plant viruses (*Han et al., 2015*). Over the past decades, many studies on virus-induced changes on the behavior of insect vectors have been published, where the underlying mechanisms were typically attributed to modifications of plant nutritive and defensive metabolites in response to virus infection (*Mauck, 2016*; *Mauck et al., 2016*). In particular, the persistently transmitted viruses (PTVs), rather than non-persistently or semi-persistently transmitted viruses, manage to cross the physical barriers, circulate within the hemolymph, and even replicate in insect vectors. Therefore, it has been speculated that PTVs utilize a direct neuromanipulation mechanism to induce behavioral changes in insect vectors; however, the molecular mechanism remains poorly understood (*Eigenbrode et al., 2018*; *Jia et al., 2018*).

Olfactory signaling, the most extensively investigated pathway, may be hijacked by plant viruses to modulate the behavior of insect vectors and their associations with host plants. For example, *Rice stripe virus* and *Southern rice black-streaked dwarf virus* were able to directly regulate the gene transcripts of odorant receptor coreceptor (ORco) and odorant-binding protein (OBP) of their insect vectors *Laodelphax striatellus* and *Sogatella furcifera*, respectively, and reversed their odorant preferences between virus infected and uninfected host plants (*Hu et al., 2019*; *Li et al., 2019*). Some rhabdoviruses have been reported to cross the salivary gland barrier through neurotrophic routes possibly due to neuroinvasion of the virus into the brain and nerve ganglia of its insect vectors (*Ammar and Nault, 1985*; *Ammar and Hogenhout, 2008*; *Chen et al., 2011*). Furthermore, studies on insect viruses demonstrate that the virus infection in the brain could impair the learning and

**eLife digest** When a plant becomes infected by a virus, its defenses get weakened, which attracts insects that are looking for an easy meal. Insects detect which plants are infected based on the color of the sickened plant and the smell of chemicals it releases. Once an insect leaves the infected plant, it may carry the virus to new plants, allowing the virus to spread.

Insects, however, prefer the easy pickings of plants that are already infected, making them less likely to spread the virus. Plant viruses have found ways to overcome this preference, but how they do this was not fully understood. Learning more about how plant viruses manipulate insects into helping them spread could allow scientists to develop new ways of protecting food crops from viral diseases.

Viruses that infect insects can trigger excessive immune system responses that damage insects' nerves and cause them to behave differently. For example, their senses may become impaired, they may move less, or be less able to remember things. This has led scientists to wonder whether plant viruses that use insects to spread might manipulate the insects' behaviors using a similar mechanism.

Now, Wang et al. have investigated whether the tomato yellow leaf curl virus –TYLCV for short – changes the behavior of whiteflies, which are known to spread the virus. The experiments showed that whiteflies typically prefer tomato plants infected with the virus, but after carrying TYLCV, they displayed equal preference for both infected and uninfected plants. Analyzing which genes were active in the whiteflies revealed that TYLCV triggers a harmful immune response which turns on genes that cause cells in the brain to die. This impairs the whiteflies' sight and sense of smell, making it harder for them to distinguish between infected and uninfected plants.

These findings suggest that the immune response triggered by the virus may be essential for the spread of TYLCV. It also identified a protein that causes the death of brain cells, leading to behavioral changes in the whiteflies. This suggests that targeting this protein, or other steps in this process, could help stop the spread of TYLCV in tomato plants.

memory function leading to sensitivity deficit and response lag in its insect host (*Han et al., 2015*). These relevant behavioral changes in infected insect hosts, including a reduction in mating and loco-motor activities, were typically defined as sickness behavior resulting from immune activation and nervous system dysfunction (*Han et al., 2015*; *Jenkins et al., 2011*; *Patot et al., 2009*). For instance, replication of the deformed wing virus (DWV, *Iflaviridae*) in the brain of *Apis mellifera* especially in the regions associated with vision and olfaction resulted in an insensitive responsiveness to food (*Iqbal and Mueller, 2007*; *Shah et al., 2009*). Likewise, some plant viruses induced a similar sickness behavior in insect vectors, possibly believing that sickness behavior could be also utilized by plant viruses (*Eigenbrode et al., 2018*; *Han et al., 2015*; *Mauck et al., 2016*).

In mammals, the sickness behavior is the result of neuroinflammation induced by activation of the innate immune system (*Godbout et al., 2005*). The toll-like receptor (TLR) and nod-like receptor (NLR) cascades that are triggered by pathogens amplify the signal of several inflammatory pro-grammed cell death formats in the central nervous system (CNS) (*Dantzer et al., 2008*; *Heneka et al., 2014*; *Heneka et al., 2018*; *LeBlanc and Saleh, 2009*; *Man et al., 2017*). With the loss of neurons, the motor and sensory deficits become irreversible in the case of permanent tissue damage to the host of pathogens (*Glass et al., 2010*; *Shattuck and Muehlenbein, 2015*). By contrast, plant viruses theoretically cause limited or no physical damage to insect vectors, but some cases of severe lesions leading to slower behavior and recognition dysfunction have been reported previously, suggesting that nerve damage and immune activation may be involved in the interaction between plant virus and its insect vector (*Ingwell et al., 2012*; *Mauck et al., 2012*; *Moreno-Delafuente et al., 2013*). Intriguingly, it has been shown that some intracellular immune responses of insect vectors, such as apoptosis and autophagy, benefit virus propagation and transmission, which is contrary to the established antiviral function in the virus-host interaction (*Chen et al., 2019*; *Chen et al., 2017*; *Huang et al., 2015*; *Wang et al., 2012*).

*Tomato yellow leaf curl virus* (TYLCV, Geminiviridae), a type member of the genus *begomovirus*, is exclusively transmitted by *Bemisia tabaci* (whitefly) in a persistent circulative manner, and causes epidemic outbreaks worldwide resulting in extensive crop yield losses (*Czosnek et al., 2017*;

*Pakkianathan et al., 2015*). TYLCV could enhance the odorant attractiveness of the plant to non-viruliferous Mediterranean (MED, also named as biotype Q) whitefly by suppressing the repellent volatile emission. After acquiring TYLCV, the whitefly displays no special preference for TYLCV-infected relative to uninfected tomato plants. This suggests that the loss of host preference in whitefly is possibly due to direct manipulation of TYLCV, but the underlying mechanism has not been fully elucidated (*Fang et al., 2013*). Therefore, we aimed to understand the neural mechanisms of viruliferous whitefly underlying this direct behavioral manipulation of host preference, as well as the resulting virus transmission. Here, we present the evidence that TYLCV induces caspase-dependent apoptotic neurodegeneration in the brain of whitefly, leading to sensory deficits in whitefly and promotes TYLCV transmission among the plant community.

## Results

### TYLCV reduces whitefly preference to virus-infected plant

Our previous field experiments have shown that non-viruliferous whitefly prefers to settle on TYLCV-infected plants, while the viruliferous whitefly does not exhibit a preference for TYLCV-infected relative to uninfected plants (*Figure 1A*). Due to the complexity of the host seeking behavior of whitefly, a short-term free-choice assay was used to compare the host preference between viruliferous and non-viruliferous whiteflies, and the result showed that whiteflies tended to prefer TYLCV-infected plants before acquiring the virus, while they exhibited an equal preference for virus-infected and uninfected plants after acquiring TYLCV (*Figure 1B*). Two sets of dual-choice experiments were conducted to separate the vision and olfaction cues from host plants, and the results revealed that a less number of viruliferous whiteflies preferred yellow light and odors released from virus-infected plants, suggesting TYLCV acquisition defected both the visual and olfactory sensitivities of whiteflies (*Figure 1C–D*).

To avoid the plant effect and directly detect the effect of virus acquisition on whiteflies, virions were purified and added to the artificial diet feed of whiteflies. Similarly, with the rearing on plants, and after the 2-day acquiring of virions, fewer whiteflies preferred virus-infected plants, suggesting that TYLCV impaired both the visual and olfactory preference of whiteflies to virus-infected plants (*Figure 1E–G*). The mTYLCV, a coat protein mutant of TYLCV in which a partial sequence of CP was substituted by the *Papaya leaf curl China virus* and hardly penetrated the whitefly gut barrier, was used to determine whether the gut barrier could prevent the impairment of host preference of viruliferous whitefly (*Guo et al., 2018*; *Wei et al., 2017*). Compared with the wild-type TYLCV virions, mTYLCV virions failed to change the host preference of whitefly in the free-choice and odorant dual-choice experiments, suggesting that a successful crossing midgut was necessary for TYLCV to modify the host preference of whitefly (*Figure 1E–G*).

### TYLCV impairs whitefly host selection ability by dysfunctioning the nervous system

In order to determine the effect of TYLCV on the host selection of whitefly, the responding time of viruliferous whitefly to plant volatiles were monitored, and it was found that the reaction of viruliferous whitefly to plant odors became slower as the feeding time increased (*Figure 1H*). It has been believed that the insensitivity of olfactory recognition was probably due to the impairment of odorant signaling in viruliferous whiteflies; however, none of the 8 *OBP*s and 12 chemosensory proteins (*CSP*s) significantly changed between viruliferous and non-viruliferous whiteflies (*Figure 1I*). Merely the *ORco*, which is an indispensable transmembrane receptor located in olfactory sensory neurons, was downregulated by TYLCV (*Figure 1I*).

To determine the difference of gene expression profile in the nervous system between viruliferous and non-viruliferous whiteflies, the transcriptome of four head samples of each treatment (eight in total) were sequenced (RNA-Seq). About 6.06 Gb data on average were generated for each sample. About 40,058,194 to 41,030,800 clean reads were obtained and mapped to the MED whitefly reference genome, and the mapping rates ranged from 69.15% to 71.55%. A total of 332 (203 upregulated, 129 downregulated) genes were differentially expressed (log2 fold change >1, adjust p-value<0.05) in the head of viruliferous whitefly compared with those of non-viruliferous whitefly (*Supplementary file 2*, *Supplementary file 3*). The enrichment analysis of KEGG pathway showed

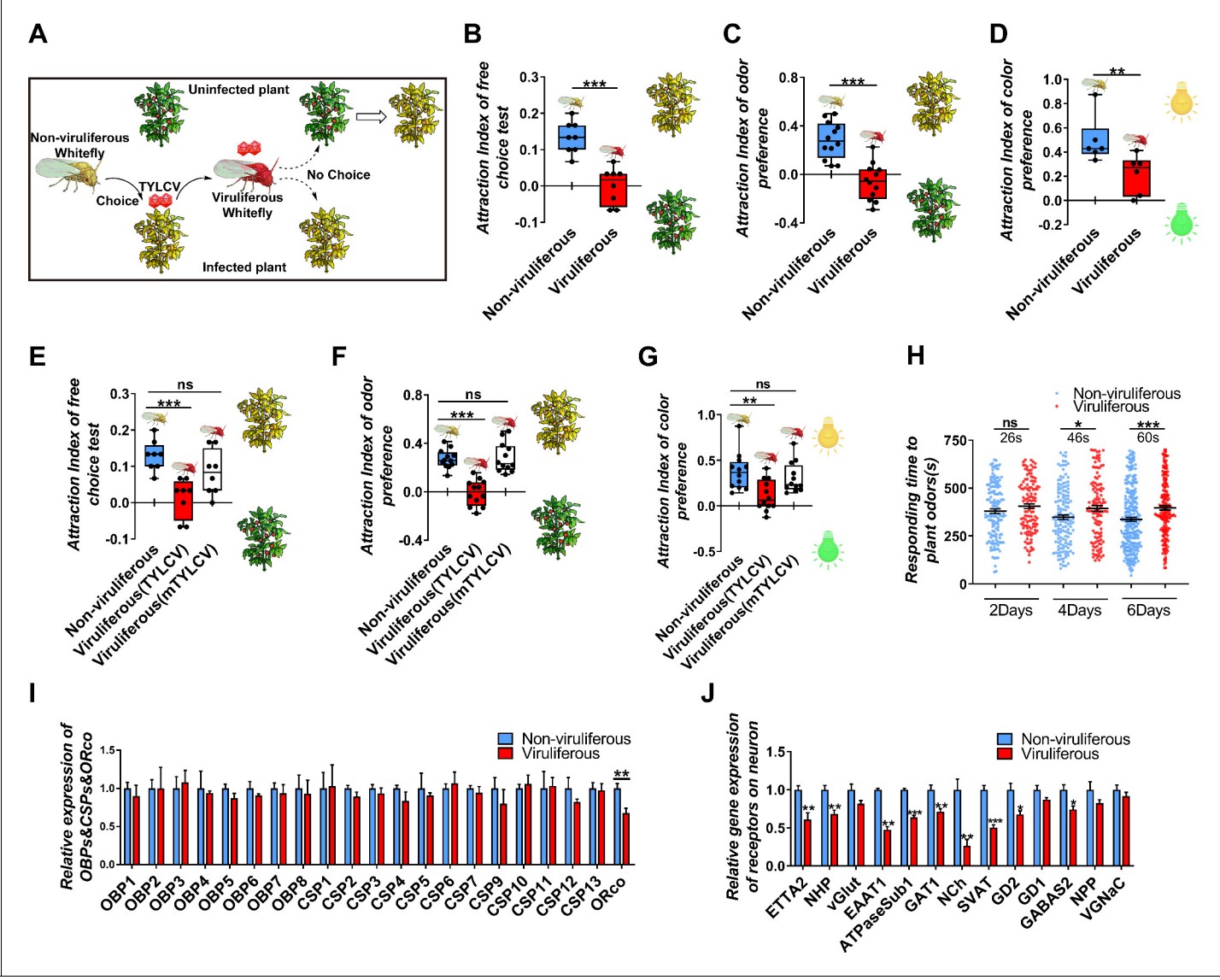

**Figure 1.** TYLCV impairs the host selectivity of whitefly between infected and uninfected host plants. (A) Whitefly preferences change after TYLCV acquisition. (B–D) Whitefly attraction index (feed on infected or uninfected plants) of (B) free-choice assay with plants, n = 8, (C) dual-choice assay with plant odors, n = 12, and (D) dual-choice assay with colors (green and yellow), n = 6. (E–G) Whitefly attraction index (feed on artificial diet with/without purified virions) of (E) free-choice assay with plants, n = 8, (F) dual-choice assay with plant odors, n = 12, (G) dual-choice assay with plant colors, n = 12, (H) Responding time to the same plant odor in 12 min-monitoring was recorded. The difference of means ± SEM is labeled. Whiteflies on infected or uninfected plants were collected separately after 2 days, 4 days, or 6 days of feeding. (I) Relative gene expression of whitefly *OBPs*, *CSPs*, and *ORco*, n = 3–5. (J) Relative gene expression of whitefly neuron membrane receptors, n = 3–5. Box plots represent the median (bold black line), quartiles (boxes), as well as the minimum and maximum (whiskers). Values in bar plots represent mean ± SEM (*p<0.05, **p<0.01, ***p<0.001). The online version of this article includes the following figure supplement(s) for figure 1:

**Figure supplement 1.** Whitefly immune system and nervous system response to TYLCV in head transcriptome analysis.

that 17 differentially expressed genes (DEGs) were enriched in the neurodegeneration pathway (*Figure 1—figure supplement 1*). The GO analysis showed that 13 DEGs were related to neurotransmitter receptors, synaptic vascular transporter proteins, and ion channel components located on neuron membranes. Furthermore, qRT-PCR assays confirmed that 10 out of 13 DEGs were downregulated by TYLCV, indicating that the TYLCV infection had substantial effects on the nervous system of whitefly (Figure J).

## TYLCV induces apoptotic neurodegeneration in the brain of whitefly

Neurodegenerative diseases, including Alzheimer's disease, Parkinson's disease, amyotrophic lateral sclerosis, and multiple sclerosis, have been well delineated in mammals, and are attributed to neuro-inflammation, along with neuronal dysfunction and death (*Glass et al., 2010*). The morphology symptom in insects exhibits vacuolar neuropathological lesions in the brain, which are caused by pathogen infection or aging (*Cao et al., 2013*; *Kounatidis et al., 2017*). In regard to vacuolar lesions in the brain of whitefly, TYLCV induced severe neurodegeneration in whitefly despite acquiring virions from plants vs. artificial diets (*Figure 2A–B*).

The progressive cell loss in specific neuronal populations during neurodegenerative disorders is mostly inducted by caspase-dependent cell death, which includes apoptosis, pyroptosis, necrosis, etc. (*Bredesen et al., 2006*; *Fan et al., 2017*; *Jellinger, 2001*). Only two caspase genes have been characterized in the genome of whitefly, namely, *BtCaspase1* and *BtCaspase3b*, which were demonstrated to be involved in cell death to initiate apoptotic response to UV stress (*Wang et al., 2018*). Acquiring virions from tomato plants and tobacco plants, and artificial diet, respectively, upregulated the gene transcripts of *BtCaspase1* and *BtCaspase3b*, both in the whole body and head of whitefly (*Figure 2C–D*, *Figure 2—figure supplement 1C*). Canonical effector caspase cleavage in model species has been well characterized and represents the initiation of apoptosis. Full-length effector caspase can be cleaved by initiator caspase into three subunits including a short prodomain, a p10 domain subunit, and a p20 domain subunit. The p20 and p10 subunits closely associate with each other to form a caspase monomer, and then, two monomers combine into an active homo-dimer that executes the apoptosis activation (*McIlwain et al., 2013*; *Riedl and Shi, 2004*). However, it remains unclear whether Caspase3b of whitefly has the same/similar function with mammals or *Drosophila melanogaster* because whitefly loses many apoptosis-related genes (*McIlwain et al., 2013*; *Nishide et al., 2019*; *Riedl and Shi, 2004*; *Zhang et al., 2014*). Using a time-course UV treatment experiment and non-reduced denaturing western blot assay with a p10 antibody, we found that inactive Caspase3b existed as 49 KDa monomer in whitefly (most effector caspases usually exist as homodimer), and the cleavage process is the same as the canonical manner (*Figure 2E*). Once apoptosis was induced by UV, cytoplasmic inactive Caspase3b rapidly decreased in 5 min, indicating cleavage initiation. Afterward, the full-length inactive Caspase3b was recovered in 15–30 min, and increased in 60–120 min, since the transcriptional expression was also induced. Cleaved p10 and monomer increased during apoptosis activation in 5–30 min, and afterward, these monomers formed the active homodimer (60–120 min) (*Figure 2E*). Together, these findings suggest that the decrease of full-length Caspase3b, and increase of p10 and monomer, as well as the increase of active homo-dimer represent the apoptosis activation in whitefly; however, they are shown in different stages of Caspase3b cleavage. This Caspase3b cleavage was also confirmed by another anti-Caspase3b p20 antibody, which displayed a similar result using p10 antibody (*Figure 2—figure supplement 2*). These results demonstrate that TYLCV virions directly induced Caspase3b cleavage in whitefly (*Figure 2F*).

TUNEL assay, which is the most frequently used method to detect cell death by labeling the end of the deoxynucleotidyl transferase dUTP, was performed to determine apoptosis in the head of vir-uliferous whitefly. Compared to the regular programmed cell death in non-viruliferous whitefly, TYLCV highly induced broad apoptosis in the brain of viruliferous whitefly, which strongly accumu-lated protein product of *BtCaspase3b* (*Figure 2G–H*). Considering the difficulty in dissecting the pure brain tissue of whitefly, FISH and immunofluorescence were used to detect virus DNA and coat protein in the head of viruliferous whitefly. The results revealed that the DNA and coat protein of TYLCV was located in the brain, eyes, and antennas of viruliferous whitefly (*Figure 2—figure supple-ment 3*). These results suggest that TYLCV induced the *BtCaspase1* and *BtCaspase3b cascade*, and caused apoptotic neurodegeneration in the brain of whitefly.

### *BtCaspase1* and *BtCaspase3b* are necessary for the TYLCV induced neurodegeneration of whitefly

The *BtCaspase1* and *BtCaspase3b* in whitefly were silenced with double-stranded RNA (*Figure 2—figure supplement 1A–B*). For *dsCaspase1* or *dsCaspase3b*-silenced whitefly, TYLCV failed to impair the preference of whitefly to TYLCV-infected plants in the free-choice and dual-choice assays (*Figure 3A–B*). The silence of *BtCaspase1* and *BtCaspase3b* at the transcription level suppressed the

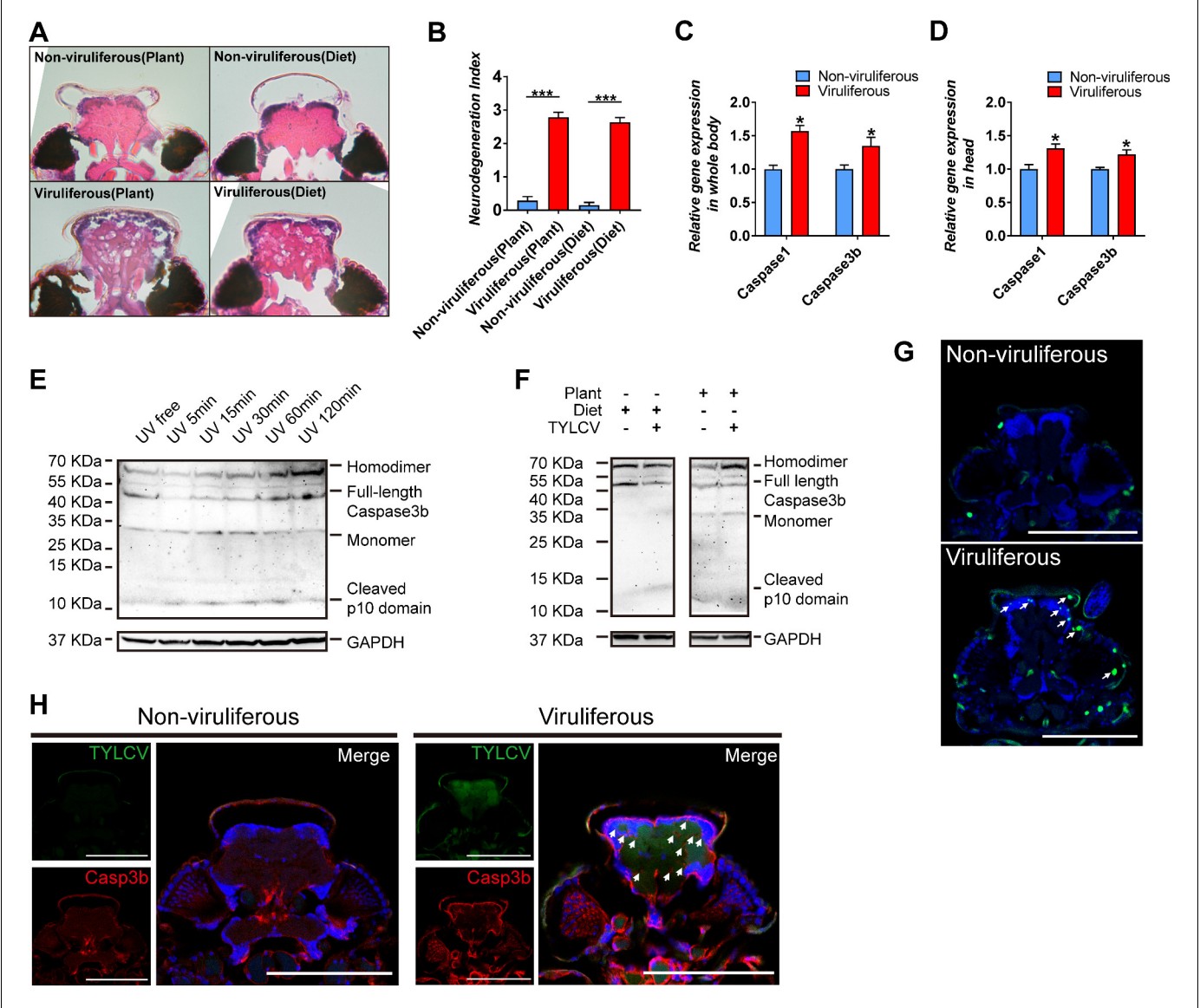

**Figure 2.** TYLCV induces apoptotic neurodegeneration in the brain of whitefly. (**A–B**) Neurodegeneration of whitefly feeding on plant diet (infected or uninfected) and artificial diet (with or without TYLCV virions) were observed (**A**) and quantified (**B**) in head sections, $n_{Uninfected\ plant}$=21, $n_{Infected\ plant}$=60, $n_{Artificial\ diet}$=21, $n_{Diet+TYLCV}$=60. (**C–D**) Both whitefly bodies with head and dissected heads were collected, and the relative gene expression of whitefly *Caspase1* and *Caspase3b* were analyzed using qRT-PCR, n = 5. (**E–F**) A 2 hr time-course UV treatment was considered as a positive control to monitor *Caspase3b* cleavage and activation by non-reduced denaturing polyacrylamide gel electrophoresis (**E**). Both purified virions and infected plants would induce *Caspase3b* cleavage and activation (**F**). All bands of each sample were imaged from the same blot, n = 3. (**G**) Whitefly head sections were fixed and labeled with terminal deoxynucleotidyl transferase-mediated dUTP nick-end labeling. Green indicates TUNEL staining of the apoptotic cells. (**H**) Head sections were labeled with anti-TYLCV CP and anti-Caspase3b antibodies. All confocal images of the head section were dissected from whiteflies feeding on artificial diet. Scale bar = 100 μm, n > 15. Values in bar plots represent mean ± SEM (*p<0.05, **p<0.01, ***p<0.001).

The online version of this article includes the following figure supplement(s) for figure 2:

**Figure supplement 1.** Relative gene expression in qRT-PCR analysis.
**Figure supplement 2.** Dynamics of Caspase3b cleavage.
**Figure supplement 3.** Localization of genome DNA and CP of TYLCV.

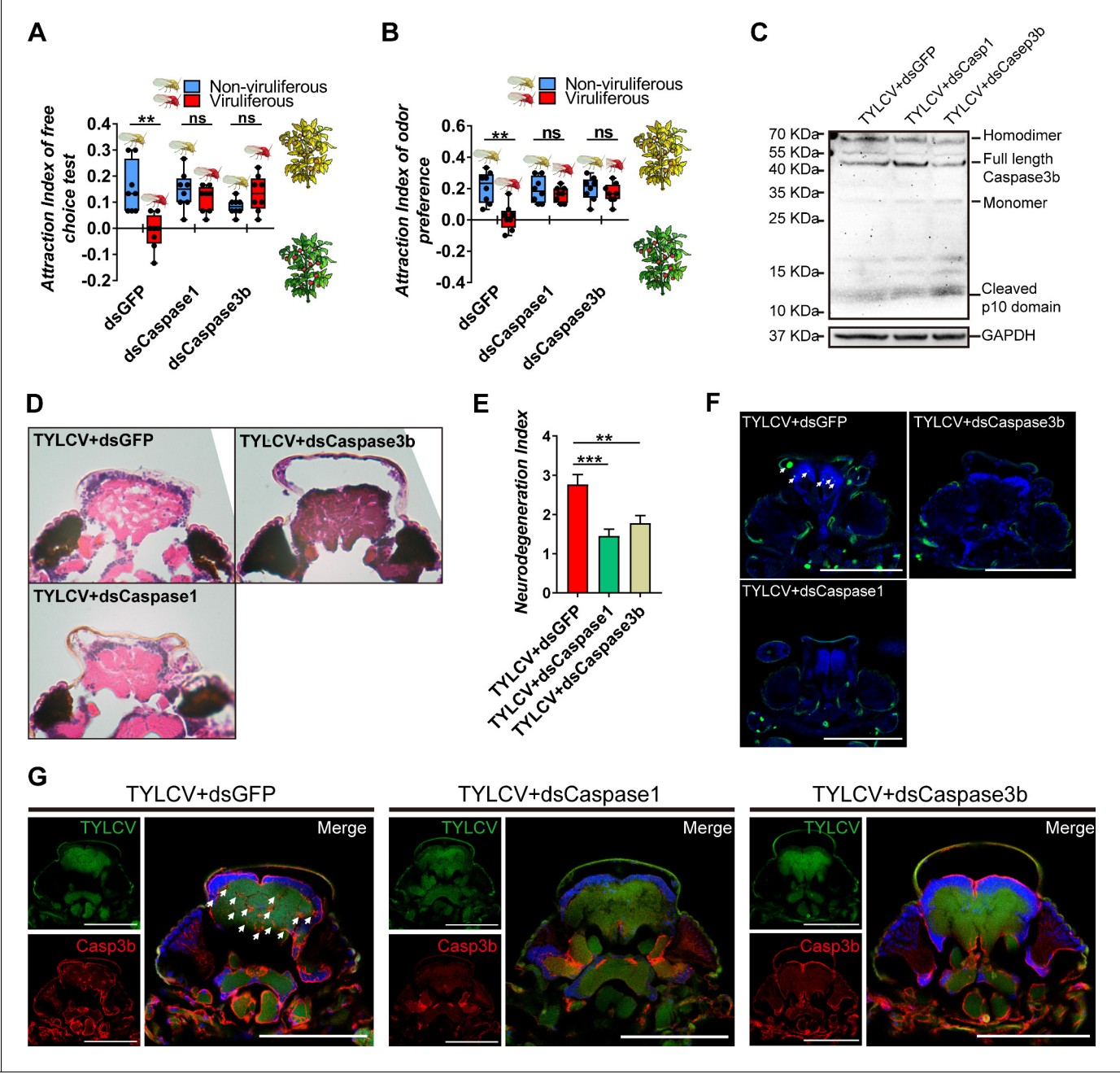

**Figure 3.** Silencing caspases alleviates virus-induced neurodegeneration. (A–B) Whitefly attraction index (fed on artificial diet with dsRNA) of (A) free-choice assay with plants, n = 8, (B) dual-choice assay with plant odors, n = 8. (C) Caspase3b of whitefly treated with virions and dsRNA was detected using western blot. (D–E) Neurodegeneration of whitefly fed with virions and dsRNA was observed (D) and quantified (E) in head sections, $n_{TYLCV+dsGFP}$=30, $n_{TYLCV+dsCaspase1}$=31, $n_{TYLCV+dsCaspase3b}$=41. (F–G) Head section images of whitefly feed with TYLCV and dsRNA. Interference with *Caspase1* and *Caspase3b* alleviates brain apoptosis in TUNEL assay (F) and Caspase3b in immunofluorescence. Scale bar = 100 μm, n > 12. Box plots represent the median (bold black line), quartiles (boxes), as well as the minimum and maximum (whiskers). Values in bar plots represent mean ± SEM (*p<0.05, **p<0.01, ***p<0.001).

cleavage of full-length *BtCaspase3b* into the homodimer at the protein level (*Figure 3C*), and subsequently ameliorated the DNA fragmentation and neurodegeneration in the brain of whitefly (*Figure 3D–F*), leading to the decreased accumulation of *BtCaspase3b* in the brain of viruliferous whitefly (*Figure 3G*). These data demonstrate that *Btcaspase3b* is required for the virus-induced neurodegeneration of whitefly, and the preference change induced by TYLCV.

### *NLRL4-Spaetzle1 and 2* signaling is involved in TYLCV-induced apoptotic neurodegeneration

In mammals, neurodegenerative disorders are usually conducted by NLRs associated with neuroinflammation (*Glass et al., 2010*). NLRs can recognize the pathogen-associated molecular patterns (PAMPs) or damage-associated molecular patterns (DAMPs), triggers the inflammasome assembly, and activates the downstream caspase cascades or cytokines release (*Glass et al., 2010*; *Guo et al., 2015*; *Heneka et al., 2018*). Four NLR-like genes had been annotated in the present transcriptome data, which were named as *NLRL*s. Furthermore, six *Spaetzle*s, which are homologous to cytokines in mammals, were also found in the transcriptome data, based on sequence similarity (*Figure 4A–B*). The qRT-PCR confirmation revealed that only *BtNLRL4* and *BtSpaetzle1 and 2* (two genes were quantified by a pair of primers due to high sequence similarity) were upregulated in viruliferous vs. non-viruliferous whitefly (*Figure 4C–D*).

Cytoplasm NLRs are structurally and functionally conserved, and widely expanded from lower metazoans to mammals, except for insects, indicating that distinct domain architectures might be specially evolved in the NLRs of insects (*Meunier and Broz, 2017*). The architecture of *BtNLRL4* was analyzed using website tools (InterPro, Prosite, and SMART), and the similarity to model species was compared using PSI-BLAST. These revealed that the leucine-rich repeat domain (LRR) of *BtNLRL4*, which generally function as the sensor of PAMPs and DAMPs, was highly similar to *NOD3* in *Homo sapiens* or *NLRC3* in *Mus musculus*. However, the NOD or PYRIN/CARD/DED domains were not identified (only predicted by SMART with a substandard significance score) (*Figure 4E*). In order to further determine the roles of *BtNLRL4* and *BtSpaetzle1 and 2*, which involved the TYLCV-induced neurodegeneration in whitefly, *BtNLRL4,* and *BtSpaetzle1 and 2* were silenced with RNA interference (*Figure 2—figure supplement 1A–B*). The TYLCV-induced neurodegeneration was ameliorated in fewer and smaller vacuolar lesions in the brain when *BtNLRL4* and *BtSpaetzle1 and 2* were silenced (*Figure 2* 4 F-G). The silencing of *BtNLRL4* and *BtSpaetzle1 and 2* also suppressed the cleavage of full-length *BtCaspase3b*, and reduced the apoptosis cells, and ameliorated the accumulation of Caspase3b in the brain of whitefly (*Figure 4H–J*). The qRT-PCR experiments revealed that TYLCV failed to upregulate *BtCaspase1* and *BtCaspase3b* when *BtNLRL4* and *BtSpaetzle1 and 2* were silenced in whitefly, respectively (*Figure 4K–L*). Likewise, TYLCV failed to suppress the host selection ability of viruliferous whitefly when *BtNLRL4* and *BtSpaetzle1 and 2* were silenced (*Figure 4M–N*). These results demonstrate that the *NLRL4-Spaetzle1 and 2* signaling was required for the TYLCV-induced caspase-dependent neurodegeneration in whitefly.

### TYLCV-induced sickness behavior promotes the virus transmission ability of whitefly

TYLCV induced a typical sickness behavior and impaired the host selection ability of viruliferous whitefly. In order to determine the effects of TYLCV-induced sickness behavior on the virus transmission ability of whitefly in the plant community, infected or uninfected plants were alternatively placed in a 1 m diameter circle, and 150 whiteflies for each treatment were released at the center for the 4 hr virus transmission experiment (*Figure 4—figure supplement 1*). A total of 16 previously uninfected plants for each treatment were marked and detected after 7 days to evaluate the virus transmission ability of viruliferous whitefly. The PCR results revealed that 15 out of 16 plants were successfully infected with TYLCV, but fewer plants were infected by viruliferous whitefly when *BtCaspase1*, *BtCaspase3b*, *BtNLRL4*, and *BtSpaetzle1 and 2* were, respectively, silenced (*Figure 4O*). These results indicate that TYLCV-induced sickness behavior promoted TYLCV transmission in the plant community.

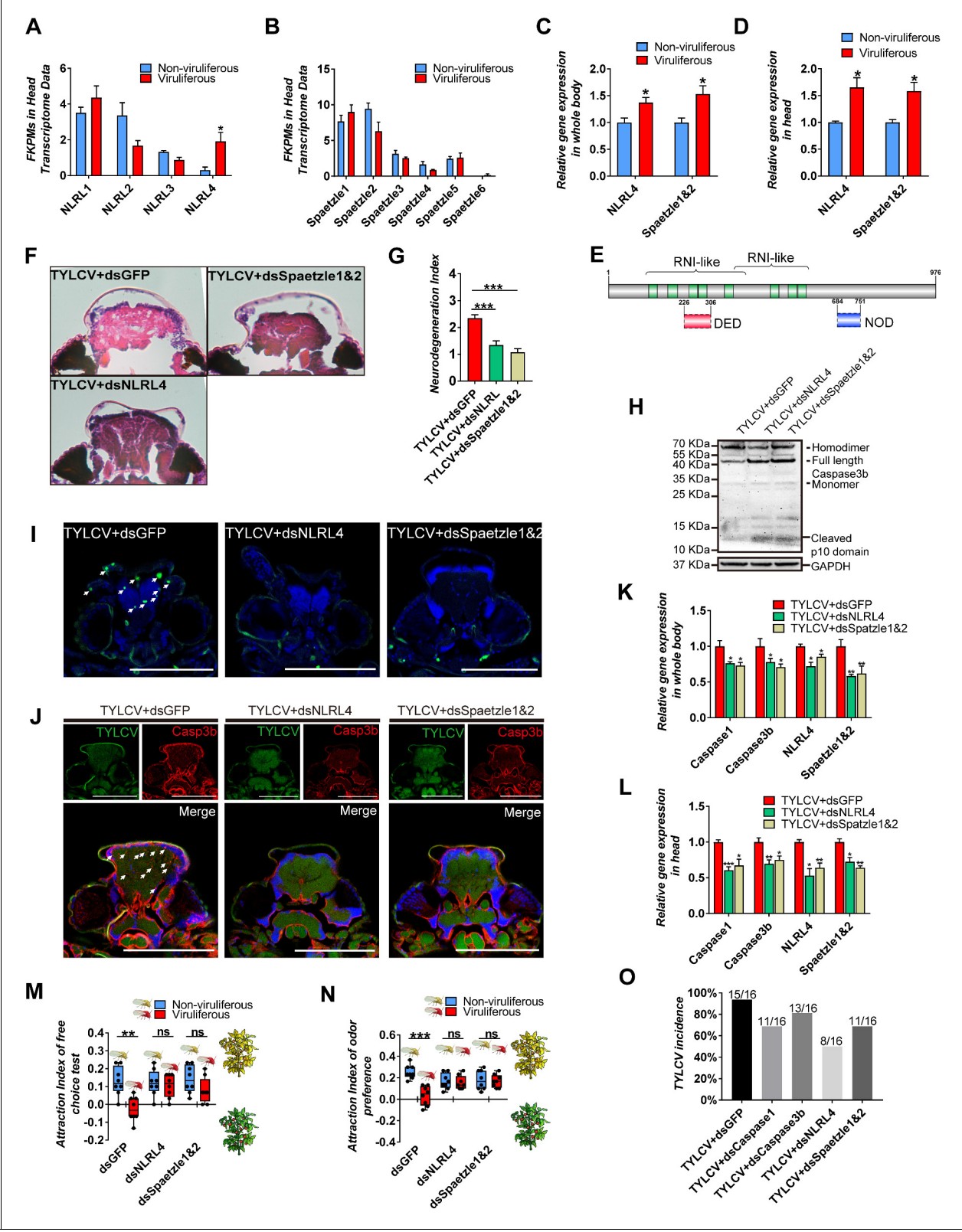

**Figure 4.** NLRL4 responses to TYLCV and induces neurodegeneration in whitefly. (A–B) FKPMs of NLRLs (A) and Spaetzles (B) in whitefly head transcriptome data. (C–D) Relative gene expression of whitefly bodies (C) or heads (D) *NLRL4* and *Spaetzle1 and 2* were analyzed using qRT-PCR, n = 5. (E) Conserved domains of NLRL4 was predicted by InterPro and SMART. Green represents leucine-rich repeats, dashed boxes represent domain scores that are less significant than the required threshold. (F–G) Neurodegeneration of whitefly fed with virions and dsRNA was observed (F) and quantified

*Figure 4 continued on next page*

*Figure 4 continued*

(**G**) in head sections, $n_{TYLCV+dsGFP}$=40, $n_{TYLCV+dsNLRL4}$=49, $n_{TYLCV+dsSpaetzle1\&2}$ = 53. (**H**) Caspase3b of whitefly treated with virions and dsRNA was detected using western blot. (**I–J**) Head section images of whitefly fed with TYLCV and dsRNA. Interference with *NLRL4* and *Spaetzle1 and 2* alleviates brain apoptosis in TUNEL assay (**F**) and Caspase3b in immunofluorescence. Scale bar = 100 μm, n > 12. (**K–L**) Interference with *NLRL4* or *Spaetzle1 and 2* suppressed caspases expression in both bodies (**K**) and heads (**L**), n = 5. (**M–N**) TYLCV cannot alter whitefly preference in (**M**) free-choice assay, n = 8, and (**N**) odorant dual-choice assay, n = 8, after interference with *NLRL4* or *Spaetzle1 and 2*. (**O**) Rescue whitefly preference impairs TYLCV transmission after interference with *Caspase1, Caspase3b, NLRL4,* or *Spaetzle1 and 2*. Box plots represent the median (bold black line), quartiles (boxes), as well as the minimum and maximum (whiskers). Values in bar plots represent mean ± SEM (*p<0.05, **p<0.01, ***p<0.001).

The online version of this article includes the following figure supplement(s) for figure 4:

**Figure supplement 1.** Virus transmission bioassay.

## Discussion

Since all known geminiviridae viruses are plant viruses, the modification of insect activities that facilitate virus transmission was largely attributed to the quality of virus-infected plants (*Eigenbrode et al., 2018*; *Liu et al., 2013*; *Moreno-Delafuente et al., 2013*). The changes in defensive metabolites and nutritive values of host plants were empirically responsible for improving the feeding efficiency and oviposition of viruliferous whitefly (*Liu et al., 2013*; *McKenzie, 2002*; *Moreno-Delafuente et al., 2013*). The disease symptoms of TYLCV-infected plants, such as yellow curl leaves and less repellent volatiles, enhanced the attractiveness to non-viruliferous whitefly to acquire TYLCV virions (*Fang et al., 2013*). It has been reasonably speculated that sustaining a strong attractiveness to viruliferous insect vectors was a disadvantage for the virus spread. In this study, we found that TYLCV was able to eliminate the unfavorable attractiveness from virus-infected plants by directly impairing the host selection ability of viruliferous whitefly. This virus-induced sickness behavior of whitefly was triggered by caspase-dependent apoptotic neurodegeneration, which was the outgrowth of the activation of innate immune response in whitefly. Our finding reveals the virus-induced immune-neuro-behavior communication in viruliferous insect vector, which promoted the virus transmission among plants.

In most cases, host preferences between viruliferous and non-viruliferous vectors are opposite, in which the non-viruliferous vector preferred virus-infected plants, while the viruliferous vector preferred uninfected plants (*Eigenbrode et al., 2018*). These opposite preferences benefit the virus spread during both the virus acquisition and transmission stages, and these have been reported to be regulated by the olfaction signaling cascades of insect vectors, including the transcriptional regulation of *OR*s and *OBP*s (*Hu et al., 2019*; *Li et al., 2019*). By contrast, our results show that viruliferous whitefly exhibits an unbiased preference between TYLCV-infected and uninfected plants, suggesting that the manipulation of TYLCV on the host preference of whitefly could differ in the modifications of olfaction signaling. Likewise, it appears that TYLCV suppressed the perception of whitefly to olfactory or visional cues from host plants, in terms of behavioral choice assays. The TYLCV-induced sensory deficits in viruliferous whitefly were similar to typical sickness behavior, even though this was a coordinated symptom of behavioral changes that responded to the infection. Furthermore, the sickness behavior was considered as an adaptive means of redirecting energy from disadvantageous behavior to an effective immune response (*Shattuck and Muehlenbein, 2015*). The insect virus-induced sickness behavior reduced the sexual activity and feeding of the insect host, which were beneficial for insect population, thereby deterring the virus spread from healthy individuals but negatively affecting the sexually transmitted pathogens (*Han et al., 2015*). In this study, viruliferous whitefly that exhibited the sickness behavior efficiently transmitted the TYLCV, when compared to those without sickness behavior, suggesting that the sickness behavior of viruliferous whitefly facilitates the spread of PTVs.

Neurodegenerative disorders accompanied by CNS cell loss are responsible for the sickness behavior in insects (*Godbout et al., 2005*; *McCusker and Kelley, 2013*; *Ransohoff, 2016*). Pathogen infection triggers the innate immune response in the CNS of *Drosophila* resulting in inflammatory signaling and neurodegeneration (*Cao et al., 2013*; *Delorme-Axford and Klionsky, 2019*; *Liu et al., 2018*). Interestingly, we found that the acquisition of PTVs induced neuropathological lesions in the brain of insect vectors, suggesting that whitefly could be directly infected by TYLCV (*Figure 2A–B*, *Figure 1—figure supplement 1*). It is noteworthy that apart from insect viruses, few

plant viruses have been found to infect or replicate in the brain of insects. However, in this study, both the coat protein and DNA of TYLCV can be stained by fluorescence confocal microscopy in the brain, eyes, and antenna of whitefly, indicating that an undiscovered nerve route of whitefly may exist for the infection of TYLCV (*Figure 2—figure supplement 3*).

Similar to mammals, caspase-dependent CNS cell death is also responsible for TYLCV-induced neurodegeneration (*Wang et al., 2018*). For virus-vector interactions, apoptosis is an effective weapon of the innate immunity to eliminate the virus infection, replication, and dissemination within the insect vector (*Chen and Wei, 2019*). These results revealed that the silencing of apoptosis-related caspases was behaviorally unfavorable to TYLCV transmission (*Figure 4O*). Likewise, for leaf-hoppers, *Rice gall dwarf virus* activates the caspase-dependent apoptosis by targeting the mitochondria, and inducing mitochondrial degeneration to promote viral infections within the insect vector (*Chen et al., 2019*). These findings suggested that vector apoptosis could be utilized by PTVs to promote the virus spread. Furthermore, compared with wild-type TYLCV, the mutant TYLCV was unable to induce the sickness behavior in whitefly, suggesting that the successful entry into the hemolymph was a preliminary step for the stimulation of whitefly innate immunity. Furthermore, the immune response was activated by TYLCV only after 12 hr of virus acquisition (*Figure 2—figure supplement 1D*), while 24–48 hr is usually required for TYLCV to cross the salivary gland barrier (*Brown and Czosnek, 2002*). It has been suggested that the initiation of the innate immune response in whitefly is essential for TYLCV to cross the physical barrier of the midgut. In addition, the immune deficiency (IMD) pathway is involved in the neurodegeneration of *D. melanogaster*, but this is exceptionally absent in some hemipteran insects, such as *Acyrthosiphon pisum*, *Diaphorina citri*, and *B. tabaci*, suggesting that whitefly quite differs from *D. melanogaster*, in terms of regulation of neurodegeneration (*Myllymäki et al., 2014*; *Nishide et al., 2019*).

NLRs, which cascade the inflammatory signaling, are required to amplify the immune activation, and causes neurodegeneration in human. Remarkably, conserved NLRs are the most abundant cytoplasmic innate immune receptors in plants and animals, other than insects. In the present study, a putative NLR, *BtNLRL4*, was found to be necessary for the TYLCV-induced neurodegeneration of whitefly, even though the architecture differed from NLRs in mammals or plants (*Figure 4F–G*). Considering the DD domain of initiator *BtCaspase1* and the putative DD/DED domain of *BtNLRL4*, the direct interaction of *BtCaspase1* and *BtNLRL4* might exist in whitefly rather than the constructing inflammasome in mammals. Spaetzles are homologous to cytokines in mammals which have been well characterized in terms of its inflammatory function. In these present results, *BtSpaetzle1 and 2* was also essential to the TYLCV-induced neurodegeneration of whitefly (*Figure 4F–G*). TYLCV was unable to upregulate the gene transcripts of *BtCaspase1* and *BtCaspase3* when *BtNLRL4* and *BtSpaetzle1 and 2* were, respectively, silenced, indicating that *BtNLRL4-BtSpaetzle1 and 2* was possibly the upstream signaling of the caspases cascade, which is consistent with the inflammatory signaling in mammals (*Glass et al., 2010*; *Heneka et al., 2018*).

The behavioral manipulation on insect vector preference is one of the most effective strategies of plant viruses to enhance their spread (*Roosien et al., 2013*). A mathematic model concluded that the host preference of insect vectors could lead to a dramatic difference in plant virus epidemic (*Roosien et al., 2013*). For single species of plants, viruliferous whitefly preferably chooses uninfected plants, which appears to be beneficial to the TYLCV spread, while the unbiased host preference is more practical and has more efficacy to the plant community, in terms of the complexity and diversity of the volatile components released from a broad range of host plants of TYLCV. Although TYLCV-induced immune responses are favorable for virus spread, it has been shown that the activation of autophagy leads to degradation of the coat protein and genome DNA of TYLCV (*Chen et al., 2017*). It has been suggested that merely the optimal degree of immune responses triggered by the virus could maximize the benefits of TYLCV in balancing the virus replication and transmission. In summary, TYLCV changes the host preference of whitefly to promote its spread by inducing the caspase-dependent apoptotic neurodegeneration in the insect vector. A deep understanding of the innate immunity and utilization of the antagonistic effect between apoptosis and autophagy in insect vectors could be an efficient approach to control the transmission and spread of PTVs in the future.

# Materials and methods

## Key resources table

| Reagent type (species) or resource | Designation | Source or reference | Identifiers | Additional information |
|---|---|---|---|---|
| Strain, strain background (*Tomato yellow leave curl virus*) | *Tomato yellow leave curl virus* isolate SH2 infectious clone | Xueping Zhou, Institute of Plant Protection, CAAS | | |
| Strain, strain background (*Tomato yellow leave curl virus*) | Mutant *Tomato yellow leave curl virus* isolate SH2 infectious clone | Xiaowei Wang, Zhejiang University | | |
| Antibody | Mouse monoclonal anti-TYLCV CP | Jianxiang Wu, Zhejiang University | | IF (1:500) |
| Antibody | Mouse monoclonal anti-GAPDH | Proteintech | Cat# 60004–1-Ig; RRID:AB_2107436 | WB (1:5000) |
| Antibody | Rabbit polyclonal anti-Caspase3b p10 | This paper | | Immunogen: YFRPKRPAIDL*C WB (1:3000) IF (1:500) |
| Antibody | Rabbit polyclonal anti-Caspase3b p20 | This paper | | Immunogen: LSQEDHSDADC WB (1:2000) |
| Antibody | Alexa 488 goat anti-mouse IgG | Abcam | Cat#ab150113; RRID:AB_2576208 | IF (1:500) |
| Antibody | Alexa 555 goat anti-rabbit IgG | Abcam | Cat#ab150078; RRID:AB_2722519 | IF (1:500) |
| Commercial assay or kit | One Step TUNEL Apoptosis Assay Kit | Beyotime | Cat#C1088 | |
| Commercial assay or kit | Absolutely RNA Nanoprep Kit | Agilent | Cat#400753 | |
| Commercial assay or kit | RoomTemp Sample Lysis Kit | Vazyme | Cat#P073 | |
| Commercial assay or kit | TRIzol Reagent | Ambion | Cat#15596018 | |
| Commercial assay or kit | FastQuant RT Kit with gDNase | Tiangen | Cat#KR106 | |
| Commercial assay or kit | PowerUp SYBR Green Master Mix | Applied Biosystems | Cat#A25742 | |
| Commercial assay or kit | T7 RiboMAX Express RNAi System | Promega | Cat#P1700 | |
| Software, algorithm | SPSS | SPSS | RRID:SCR_002865 | |
| Software, algorithm | GraphPad Prism software | GraphPad Prism (https://graphpad.com) | RRID:SCR_015807 | |

## Insect rearing

The Mediterranean (MED)/Q whitefly (mtCOI GenBank accession no: GQ371165) of the *Bemisia tabaci* species complex were reared on cotton plants (*Gossypium hirsutum* cv Guo-Shen 7886) placed in insect-proof cages between 26°C and 28°C, with a photoperiod of 16:8 hr (light/dark). Adult male or female whiteflies were randomly obtained using a sucking device for the experiments.

## Tomato yellow leaf curl virus

The infectious clone of TYLCV isolate SH2 (GenBank accession no: AM282874) was provided by Professor Xueping Zhou (State Key Laboratory for Biology of Plant Diseases and Insect Pests, Institute of Plant Protection, Chinese Academy of Agricultural Sciences). The infectious clone of mutant TYLCV was provided by Xiaowei Wang (Institute of Insect Sciences, Zhejiang University).

## Plants

Tomato plants (*Solanum lycopersicum* cv Moneymaker) were used as the natural host for both TYLCV and whitefly in behavioral assays. Virus-infected *Nicotiana benthamiana* was used to purify the TYLCV virions. All plants were reared at 26–28°C, with 60% relative humidity and a photoperiod of 16:8 hr (light/dark). The infectious clone was inoculated into plants at the 3–4 true leaves stage, and plants with both obvious symptoms and a positive result in the PCR analysis were used in the experiments as TYLCV infected plants.

## Behavioral assays

A total of 30 whiteflies from the same treatment were collected in a clean pipet tip as one biological replicate for all behavioral assays. The free-choice assay was performed in an insect-proof cage (40 × 40 × 40 cm³), with two TYLCV infected and two uninfected tomato plants placed diagonally. Then, the number of whiteflies on each plant was counted after a 10 min choosing. Any whitefly that settled on the ground or hung on the cage was recorded as 'no choice'.

For the olfactory related dual-choice assay, two hermetical-sealed glass chambers (40 cm in height, 23 cm in diameter) that contained the test plants were connected to two branch arms of a glass Y-tube olfactometer (24 cm in length for each arm), respectively. A purified airflow was equally and continually pumped from the glass chamber to the olfactometer at 300 ml/min. For the visual-related dual-choice assay, two branch arms of the olfactometer were connected to the same glass chamber and equally illuminated by green and yellow light, respectively. For the responding test, one branch arm of the Y-tube was connected to the glass chamber that contained the tomato plant, and another branch arm was connected to the clean airflow. The Y-tube olfactometer was rotated at 180° after each replicate to avoid positional bias, and all tests were conducted between 15:00 and 20:00, in case of the circadian difference. Whiteflies were released in the main olfactometer arm for up to 10 min. A choice for one of the two branch arms was considered as valid when the whitefly moved >5 cm onto either arm, and stayed in that arm for at least 15 s. This experiment design has been modified from Fereres' and Li's works (*Fereres et al., 2016*; *Li et al., 2014*).

Whitefly preference was quantified with an attraction index (AI), calculated as: AI = ($V$ - $N$)/30, where $V$ is the number of viruliferous whiteflies, $N$ is the number of non-viruliferous whiteflies, and 30 is the sum of whiteflies was used in one test. The AI calculation method was also performed in previous study (*Keesey et al., 2017*).

## Virions purification

TYLCV particles were isolated from young leaves of *Nicotiana benthamiana* plants after post-inoculation 3 weeks of the TYLCV infectious clone. The brief protocol was previously described and modified in the present experiment (*Pakkianathan et al., 2015*). One gram of fresh weight leaf tissue was homogenized in 2.4 ml of ice-cold lysis buffer (pH 8.0, 100 mM trisodium citrate, 18.5 mM ascorbic acid, 60 mM sodium sulfite, 5 mM EDTA, and 1% [wt/vol] β-mercaptoethanol) and produced in 2.5% (vol/vol) Triton X-100, stirred overnight, filtered through four layers of cheesecloth, and clarified by 10 min centrifugation at 8000 g. The supernatant was filtered through 0.2 μm of Supor membrane Non-Pyrogenic (PALL) and centrifuged for 3 hr at 90,000 g in a SW41Ti rotor with a coulter optima XPN-80 ultracentrifuge (Beckman). The pellet was resuspended in buffered (pH 8.0) CEM buffer (10 mM trisodium citrate, 1 mM EDTA, and 0.1% β-mercaptoethanol), and loaded onto a 10.5 ml linear

10–50% sucrose gradient in CEM buffer. The sucrose gradient was fractionated (1 ml per fraction) after 3 hr of centrifugation at 90,000 g. The positive fractions (determined by PCR) were diluted with CEM buffer, and centrifuged for 3 hr at 90,000 g, and the final pellet was suspended in 15% sucrose in CEM buffer (without β-mercaptoethanol). The presence of viral particles was confirmed by staining with 2% phosphotungstic acid, and observed using a Tecnai G2 F20 TWIN transmission electron microscope. Then, about 12 g of fresh weight leaves was finally suspended in 1 ml of CEM buffer for whitefly artificial diet feeding.

## mRNA sequencing and data analysis

A total of eight whitefly samples were respectively collected from TYLCV infected or uninfected tomato plants for sequencing, and each sample contained approximately 300 dissected heads. The RNA was extracted using an Absolutely RNA Nanoprep Kit (Agilent). The concentration and quality of the total RNA were determined using a NanoDrop spectrophotometer (Thermo) and by gel electrophoresis. Beads that contained oligo (dT) were used to isolate the poly(A) mRNA from the total RNA. The purified mRNA was fragmented in the fragmentation buffer, and used as templates to synthesize the first-strand cDNA. Then, the second-strand cDNA was synthesized using a buffer, dNTPs, RNase H, and DNA polymerase I. The short fragments with additional 'A' base were ligated to the Illumina sequencing adaptors. The selected size DNA fragment was amplified by PCR, and sequenced on an Illumina HiSeq 2000 sequencing machine. The dirty raw reads were removed before analyzing data. Then, the resulting reads were aligned to the MED whitefly reference genome (http://gigadb.org/dataset/100286) from the Giga Database, and the fragments per kilobase of transcript per million fragments mapped (FPKM) were estimated. DEseq2 was used to filtrate the differentially expressed genes (DEGs). Then, the enrichment analysis of KEGG was performed to identify the regulation pathways represented by these DEGs. The transcriptome raw data has been released already with ID: PRJNA606896.

## Histology and neurodegeneration score

Whiteflies were collected and placed in 4% paraformaldehyde fixative overnight at 4℃, washed in 70% ethanol and processed into the Tissue-Tek O.T.C. Compound (Sakura). After freezing at −20℃, the embedded whiteflies were sectioned at 10 µm using a Leica CM1950 freezing microtome and stained with H & E according to the standard protocol. Images were taken using a Nikon light microscope, equipped with a DS-Fi1c camera (Nikon), and the images were generated using the NIS-Element D software (Nikon). The appearance of vacuolar lesions in the brain neuropil was the typical symptom of neurodegeneration, and six levels of neurodegeneration (0, 1, 2, 3, 4, and 5) were defined for quantification in previous research (Cao et al., 2013). The same standard was applied to quantify the neurodegeneration of whiteflies.

## PCR and quantitative real-time PCR

The total DNA of plants or whiteflies was extracted using the RoomTemp Sample Lysis Kit (Vazyme), according to manufacturer's protocol, and a 412 bp fragment of TYLCV was amplified with the standard PCR protocol using 2 × Taq PCR MasterMix (Tiangen). The primers, which were named V61 and C473, as previously described (Atzmon et al., 1998). The total RNA of the whitefly samples for the RT-qPCR analysis were extracted by TRIzol Reagent (Ambion), and reverse transcribed using the FastQuant RT Kit with gDNase (Tiangen). The RT-qPCR reactions were carried out on the PikoReal 96 Real-Time PCR System (Thermo) using the PowerUp SYBR Green Master Mix (Applied Biosystems). Three technical replicates were applied for each biological replicate. The data was analyzed by relative quantification with the $2^{-\triangle\triangle CT}$ method. For the RT-qPCR, the sequences and partial primers of caspases, OBPs and CSPs were obtained from previous studies (Wang et al., 2017; Wang et al., 2018), while others were obtained from the MED whitefly genome data (Xie et al., 2017), and primers were designed by Primer Premier 6. Actin was used as the housekeeping gene in the experiments. The availability of each pair of primers had been tested in the preliminary experiments. The oligonucleotides are listed in Supplementary file 1. except CSP9. CSP9 was not detected in our study.

## Western blot analysis

A total of 100 whole whiteflies were pooled per condition per experiment and lysed in 150 µl of RIPA buffer (CST) supplemented with a protease inhibitor cocktail (CST). Protein samples were separated by non-reduced denaturing polyacrylamide gel electrophoresis with 8–20% or 15% precast Tris-glycine gel (EZBiolab) and transferred onto 0.22 µm polyvinylidene difluoride membranes (Millipore). Then, these membranes were blocked with SuperBlock T20 blocking buffer (Pierce), and incubated with the primary antibody. After incubation with the secondary antibody (CST), the signals were visualized using the SuperSignal West Pico PLUS Chemiluminescent Substrate (Pierce). Using the synthetic peptides YFRPKRPAIDL*C (Caspase3b 427-437aa, p10 domain) or LSQEDHSDADC (Caspase3b 252-262aa, p20 domain) as the immunogen, the rabbit-anti Caspase3b polyclonal antibody was prepared by Beijing Genomics Institute (BGI). The commercial primary antibody mouse anti-GAPDH (60004–1-Ig) was purchased from Proteintech, and the commercial anti-mouse (ab6789) and anti-rabbit (ab6721) were purchased from Abcam. After the Caspase3b (dilution 1:1,000) signals were visualized, the same blot was stripped by Restore Western Blot stripping buffer (Pierce), and incubated with anti-GAPDH (dilution 1:10,000) after blocking, in order to ensure that these two different signals (Caspase3b and GAPDH) came from the same protein sample. Three independent experiments were performed.

## RNA interference

The dsRNA was synthesized using the T7 RiboMAX Express RNAi System (Promega), according to the manufacturer's protocol. Approximately 150 adult whiteflies were placed in 30 mm in diameter, by 60 mm in height cylindrical dark containers. Each container provided a 500 µl diet with 400 µl of 15% sucrose, 50 µl purified virions (or CEM buffer as control), and 50 µl 10 µg/µl dsRNA (dsGFP as control). All RNA and protein samples were collected after a 48 hr feeding. Each treatment was replicated for at least three times.

## Immunofluorescence and confocal microscopy

The frozen sections were rinsed for three times in TBST (TBS with 0.05% Tween-20) and blocked with SuperBlock T20 (Pierce). Then, the samples were incubated with the primary antibody (anti-Caspase3b, 1:500; anti-TYLCV CP, 1:500) overnight at 4°C, rinsed three times in TBST, and incubated with the secondary antibody (1:1000) at room temperature for 2 hr. Negative controls had been performed in each independent experiment. The monoclonal antibody mouse anti-TYLCV CP was kindly provided by Professor Jianxiang Wu (Institute of Biotechnology, Zhejiang University). The anti-mouse conjugate Alexa 488 (ab150113) and anti-rabbit conjugate Alexa 555 (ab150078) were purchased from Abcam. The samples were rinsed for three times in TBST, and mounted in Fluoroshield Mounting Medium with DAPI (Abcam). Sections were imaged using a Zeiss LSM710 confocal microscope.

## TUNEL assay

The apoptotic cell death of the frozen sections with different treatments were analyzed using a One Step TUNEL Apoptosis Assay Kit (Beyotime). The sections were rinsed twice in PBS, and incubated with PBST (PBS with 0.5% Triton-X) for 5 min at room temperature. Then, these sections were rinsed twice in PBS, and incubated with the TUNEL mixture (Enzyme Solution: Label Solution = 1:9) for 1 hr at 37°C. After rinsing thrice in PBS, these sections were mounted and imaged using a microscope.

## Fluorescence in situ hybridization (FISH)

The whole whitefly was fixed in Carnoy's fixative (chloroform-ethanol-glacial acetic acid [6:3:1, vol/vol]) overnight at 4°C, and rinsed for three times in TBS. After washing by TBST (TBS with 0.2%Triton-X) for 10 min, the whitefly was rinsed for three times in hybridization buffer (20 mM Tris-HCl, pH 8.0, 0.9 M NaCl, 0.01% [wt/vol] sodium dodecyl sulfate, 30% [vol/vol] formamide) for pre-hybrid (without the probe). Then, 10 pmol of the fluorescent DNA probe (conjugated with Cy5) was added into 500 µl of hybridization buffer, and the whitefly was hybridized overnight at room temperature in the dark. Afterward, the hybridized whitefly was rinsed for three times in TBS, and mounted before imaging by microcopy. The probe was described in a previous study and is listed in *Supplementary file 1* Table S1 (*Pakkianathan et al., 2015*).

## Statistical analysis

Statistical analyses were performed using SPSS (Chicago, IL). The Wilk-Shapiro test was used to determine the normality of each data set. Normally distributed data were then analyzed using two-tailed, paired $t$-test. Nonparametric distributed data were assessed using Mann–Whitney test. An asterisk denotes statistical significance between two groups (*$p<0.05$, **$p<0.01$, ***$p<0.001$).

## Acknowledgements

This project was supported by the National Key Research and Development Plan (2017YFD0200400) and the Strategic Priority Research Program of the Chinese Academy of Sciences (XDB11050400).

## Additional information

### Funding

| Funder | Grant reference number | Author |
|---|---|---|
| Ministry of Science and Technology of the People's Republic of China | the National Key R&D Program of China (no. 2017YFD0200400) | Feng Ge Yucheng Sun |
| Chinese Academy of Sciences | Strategic Priority Research Program (no. XDB11050400) | Feng Ge Yucheng Sun |

The funders had no role in study design, data collection and interpretation, or the decision to submit the work for publication.

### Author contributions

Shifan Wang, Conceptualization, Data curation, Formal analysis, Investigation, Visualization, Methodology, Writing - original draft; Huijuan Guo, Conceptualization, Methodology, Writing - review and editing; Feng Ge, Yucheng Sun, Conceptualization, Resources, Supervision, Funding acquisition, Project administration, Writing - review and editing

### Author ORCIDs

Shifan Wang (iD) https://orcid.org/0000-0003-0640-0016
Huijuan Guo (iD) http://orcid.org/0000-0002-4432-6446
Yucheng Sun (iD) https://orcid.org/0000-0003-2353-0218

### Decision letter and Author response

Decision letter https://doi.org/10.7554/eLife.56168.sa1
Author response https://doi.org/10.7554/eLife.56168.sa2

## Additional files

### Supplementary files

• Supplementary file 1. KEGG and GO enrichment of head transcriptome, western blot labeled by a Caspase3b p20 antibody, immunofluorescence and FISH of TYLCV, RNA interference efficiency.

• Supplementary file 2. The list of differentially expressed genes (DEGs).

• Supplementary file 3. The list of all gene expression (FPKM).

• Transparent reporting form

### Data availability

Sequencing data have been deposited in SRA under accession ID PRJNA606896.

The following dataset was generated:

**Database and**

| Author(s) | Year | Dataset title | Dataset URL | Identifier |
|-----------|------|---------------|-------------|------------|
| Wang S | 2019 | MED whitefly head transcriptome | https://www.ncbi.nlm.nih.gov/sra/PRJNA606896 | NCBI Sequence Read Archive, PRJNA606896 |

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
