## [Decision Letter]

**Acceptance summary:**

The study presents interesting and informative data detailing the impacts of TYLCV infection on the brains of a whitefly vector and the potential correlations between viral infection, neuronal apoptosis, and insect behavior. The study allows for further study of this agriculturally important vector. The study examines the molecular mechanisms that underlie behavior in whiteflies that eventually aid a plant virus in its spread from infected plant hosts to healthy plant hosts. The authors show that there is neurodegeneration in the whitefly brain that could result in the infected fly preferring uninfected plant hosts.

**Decision letter after peer review:**

Thank you for submitting your article "Apoptotic neurodegeneration in whitefly promotes spread of TYLCV" for consideration by *eLife*. Your article has been reviewed by two peer reviewers, and the evaluation has been overseen by a Reviewing Editor and K VijayRaghavan as the Senior Editor. The reviewers have opted to remain anonymous.

The reviewers have discussed the reviews with one another and the Reviewing Editor has drafted this decision to help you prepare a revised submission.

Summary:

The current paper entitled "Apoptotic neurodegeneration in whitefly promotes the spread of TYLCV" presents interesting and informative data detailing the impacts of TYLCV infection on the brains of a whitefly vector and the potential correlations between viral infection, neuronal apoptosis, and insect behavior. The progression of experiments presented in this paper is good and quite clear. This work is novel and has a lot of merits as it looks at the molecular mechanisms that underlie sickness behavior in whiteflies that eventually aid a plant virus in its spread from infected plant hosts to healthy plant hosts. The authors show that there is neurodegeneration of the whitefly brain that seems to be causing this vector to not show a preference for infected plants anymore (like uninfected vectors). To reveal the mechanisms involved in and as a result of the neurodegeneration-driven change in whitefly behavior, the authors take an integrative approach by combining behavioral assays, with histology (i.e., histology, TUNEL, FISH and immunofluorescence), gene expression analyses, and functional genetics analyses. However, the overarching goal/rationale of the paper is not particularly well-defined in the Introduction, and the text at present lacks a clear flow making the data a bit difficult to interpret at first.

Essential revisions:

1) There is a good amount of background in the Introduction, but it was a bit difficult to follow in terms of flow leading up to the rationale for the paper and the description of the findings. It would help add to the strength of the paper to tighten the Introduction and ensure the justification for the present study is clear up front.

2) Figures within Figure 1 appear to be incorrectly referenced in the text. Figure 1E-F is referenced and said to show that TYLCV infection (acquired through diet) impairs visual and olfactory preference of insect vectors, similar to what is observed in Figure 1B-D (subsection “TYLCV reduces whitefly preference to virus-infected plant”). However, Figure 1E demonstrates that the reaction of infected whiteflies to plant odors slows as feeding time increases (subsection “TYLCV impairs whitefly host selection ability by dysfunctioning the nervous system” – this data appears to be incorrectly referenced as Figure 1G in the text).

3) Figure 1F and G demonstrate that whiteflies artificially infected with TYLCV through the diet demonstrate similar behavioral shifts as to what is observed in vectors that were infected with TYLCV from contact with infected plants…this is determined by the free-choice and odor preference behavioral tests (e.g., Figure 1B-C). However, Figure 1 does not include data on whether or not there are similar shifts in visual preference in artificially infected vectors as is observed in whiteflies that are infected via contact with plants. Were experiments performed looking at the effects of diet acquired TYLCV infection on visual preference as well? Or only free-choice and odor? If it wasn't assessed, why not? Also, data on visual preferences of vectors infected with mutant virus (mTYLCV) were not shown. Were these experiments performed?

4) The Western blots and demonstration of Caspase activation are not clear. In the text, it is stated that the functional caspase unit is a homodimer, and that a reduction in the presence of full length Caspase3b and an increase in the presence of the homodimer are indicative of apoptosis in whiteflies (subsection “TYLCV induces apoptotic neurodegeneration in the brain of whitefly”). While elevated immunoreactivity of the homodimer is evident, it appears marginal, and changes in full length Caspase 3b are more difficult to discern. It may help to quantitate these changes to aid in interpretation.

5) For experiments using the mutant virus mTYLCV, was it confirmed that the virus does not cross the gut barrier? For example, were viral genomes quantitated in different locations throughout the body (e.g., the head versus the gut) and compared between whiteflies infected with wildtype virus versus those infected with mutant virus? Are there any differences in viral replication between the two strains? If this is known, or if it has been demonstrated before that this specific mutation in the coat protein of TYLCV inhibits virus from crossing the gut barrier, adding the appropriate references would suffice.

6)The functional genetics analyses combined with (fluorescent) histological assessments look very convincing. We are assuming that the RT-QPCR data does too, though there is not enough information given about the reference genes used, their stability across samples and the efficiency of the primers. This makes it impossible to assess if the RT-QPCR data is valid because these are all things that influence the data presented and if the ΔCT method can be even applied. We are also not sure why the authors log transformed their RT-QPCR data. Is this perhaps a form of p-hacking? If the RT-QPCR data has been dealt with accordingly (which we assume for now) this data is nicely in line with the histology, behavior and functional gene assays.

7) The other aspect that is difficult to interpret were the immunodetection protein gels that would confirm caspase activity related to apoptosis. Those gels do not look convincing but it might be partly due to the explanation that the authors provided, which could be better stated.

8) The authors have not provided enough data and information with regards to their RNASeq experiment. Read size, number of reads, % coverage, number of DEGs and a supplementary file with DEGs does not seem to have been given. The authors should at least report their RNASeq stats. However, they have also not fully explored the dataset it seems, or given the reader more of an interpretation making this just "a list of genes" which is so often said of transcriptomics datasets while a more extensive analysis would make them useful for researchers besides those publishing them. At the moment, the RNASeq data feels more like an add on that doesn't have much to do with the manuscript because the authors mostly go by RT-QPCR finds. The work is pretty complete without the RNASeq data but it seems like this powerful data could be exploited more by doing for instance a WGCNA or PCA + loading plot analysis to get more from it.

---

## [Author Response]

Essential revisions:1) There is a good amount of background in the Introduction, but it was a bit difficult to follow in terms of flow leading up to the rationale for the paper and the description of the findings. It would help add to the strength of the paper to tighten the Introduction and ensure the justification for the present study is clear up front.

As suggested, the whole Introduction section was shortened from 1044 words to 835 words, and especially for the parts of overarching goal and description of the findings. Rewritten parts can be found in the Abstract and throughout the Introduction.

2) Figures within Figure 1 appear to be incorrectly referenced in the text. Figure 1E-F is referenced and said to show that TYLCV infection (acquired through diet) impairs visual and olfactory preference of insect vectors, similar to what is observed in Figure 1B-D (subsection “TYLCV reduces whitefly preference to virus-infected plant”). However, Figure 1E demonstrates that the reaction of infected whiteflies to plant odors slows as feeding time increases (subsection “TYLCV impairs whitefly host selection ability by dysfunctioning the nervous system”, – this data appears to be incorrectly referenced as Figure 1G in the text).

Sorry for of the mislabeling. Figure 1 has been replaced by a new version that contains the result of visual preference (acquired through diet, Figure 1G), and removes the data of responding experiment to Figure 1H. The relevant descriptions within manuscript have also been corrected. Here refers to: “Figure 1E-G” (subsection “TYLCV reduces whitefly preference to virus-infected plant”, last paragraph), “Figure 1H” and , “Figure 1I” (subsection “TYLCV impairs whitefly host selection ability by dysfunctioning the nervous system”, first paragraph), “Figure 1J”(see the last paragraph of the aforementioned subsection), and the figure legend.

3) Figure 1F and G demonstrate that whiteflies artificially infected with TYLCV through the diet demonstrate similar behavioral shifts as to what is observed in vectors that were infected with TYLCV from contact with infected plants…this is determined by the free-choice and odor preference behavioral tests (e.g., Figure 1B-C). However, Figure 1 does not include data on whether or not there are similar shifts in visual preference in artificially infected vectors as is observed in whiteflies that are infected via contact with plants. Were experiments performed looking at the effects of diet acquired TYLCV infection on visual preference as well? Or only free-choice and odor? If it wasn't assessed, why not? Also, data on visual preferences of vectors infected with mutant virus (mTYLCV) were not shown. Were these experiments performed?

We did perform the visual preference experiments with both plant and artificial diet treatment, and the result of artificial diet treatment presented as Figure 1G in our revised manuscript which was consistent with the result of odor preference. We did not further determine the functions of caspases, NLRL4, and spaetzle1&2 of whitefly by visual preference experiments.

The first reason is that we preferred olfactory preference as our main phenotype rather than visual preference in current study since the non-viruliferous whitefly preference to infected plant were dominantly affected by plant volatiles. A previous study found that TYLCV infection can suppress the repellent terpenes emission and increased plant attractiveness to whitefly (Fang et al., 2013). Besides, despite yellow leaf is one of the typical symptoms of TYLCV infection, it is difficult to quantify such changes during the virus infection and replication process. The second reason is that the visual experiments conducted with artificial yellow and green illuminations may not fully represent the natural visual difference between infected and uninfected host plants (i.e. a mild infection symptom might not be distinguishable enough). We expect our newly added Figure 1G is acceptable for this concern.

4) The Western blots and demonstration of Caspase activation are not clear. In the text, it is stated that the functional caspase unit is a homodimer, and that a reduction in the presence of full length Caspase3b and an increase in the presence of the homodimer are indicative of apoptosis in whiteflies (subsection “TYLCV induces apoptotic neurodegeneration in the brain of whitefly”). While elevated immunoreactivity of the homodimer is evident, it appears marginal, and changes in full length Caspase 3b are more difficult to discern. It may help to quantitate these changes to aid in interpretation.

Sorry for not explaining well the dynamics of Caspase activation. Reviewer #2 also mentioned that the result of Western Blots made reader confused. All the blotting in our study were performed in a non-reduced denaturing manner (already mentioned in the Materials and methods), which could not dissociate the polymer to monomer and allowed us to monitor the cleavage of caspase once upon apoptosis initiation.

Canonical effector caspase cleavage in model species has been well characterized and represents the initiation of apoptosis. Full length effector caspase can be cleaved by initiator caspase into three subunits including a short prodomain, a p10 domain subunit, and a p20 domain subunit. The p20 and p10 subunits closely associate with each other to form a caspase monomer, and then, two monomers combine into an active homodimer that execute the apoptosis activation (McIlwain et al., 2013; Riedl and Shi, 2004). It however remains unclear that Caspase3b of whitefly has the same/similar function with the mammals or *Drosophila melanogaster*, because whitefly loses many apoptosis-related genes.

**Author response image 1. sa2fig1:** 

In our study, inactive caspase3b existed as 49KDa monomer in whitefly (most effector caspases usually exist as homodimer), and the cleavage process is the same as the canonical manner (Figure 2E). Once apoptosis was induced by UV, cytoplasmic inactive caspase3b rapidly decreased in 5min, indicating the cleavage was initiated. Afterwards, the full-length inactive caspase3b recovered in 15-30min, and increased in 60-120min, because the transcriptional expression was induced. Cleaved p10 and monomer increased during apoptosis activation in 5-30min, and afterward, these monomers formed the active homodimer (60-120min) (Figure 2E). In our experiments, each protein sample was extracted from a mixture bio-sample which consistent of more than 100 whiteflies, and the status of virions acquisition of each individual whitefly could not be accurately sampled. In other words, the decrease of full length caspase3b, and increases of p10 and monomer, as well as the increases of active homodimer represented the apoptosis activation in whitefly respectively, but they displayed in different stages of caspase3b cleavage. Therefore, we need to determine the caspase3b cleavage by a time-course experiment.The revised paragraph can be found in the subsection “TYLCV induces apoptotic neurodegeneration in the brain of whitefly”: “Canonical effector caspase cleavage in model species has been well characterized and represents the initiation of apoptosis. […] These results demonstrated that TYLCV virions directly induced caspase3b cleavage in whitefly (Figure 2F).”

We also noticed that our most concerning data is Figure 2F (diet feeding treatment), and here, another two replicates (different bio-samples) of this treatment were attached. Both p10 and monomer increased in replicate 2, while only monomer increased in replicate 3 and no significant change was shown among homodimer bands. Considering the individual variations in virion acquisition, the quantification of the mixture samples of whiteflies may not be helpful to detect the differences.

5) For experiments using the mutant virus mTYLCV, was it confirmed that the virus does not cross the gut barrier? For example, were viral genomes quantitated in different locations throughout the body (e.g., the head versus the gut) and compared between whiteflies infected with wildtype virus versus those infected with mutant virus? Are there any differences in viral replication between the two strains? If this is known, or if it has been demonstrated before that this specific mutation in the coat protein of TYLCV inhibits virus from crossing the gut barrier, adding the appropriate references would suffice.

This mutant virus mTYLCV was constructed through overlap extension PCR and was already used in previous study (Wei et al., 2017). A 141 aa fragment (aa 82-222) of TYLCV CP region was replaced with a 140 aa fragment (aa 82-221) of the PaLCuCNV CP region (Wei et al., 2017). Guo et al. found that MED whitefly (the same biotype used in our experiments) had similar TYLCV acquisition ability with MEAM1 whitefly (Figure 1A in Guo et al., 2018), but acquired less mTYLCV than MEAM1 (Figure 7B in Guo et al., 2018). TYLCV was detected in both midgut (MG) and hemolymph (HL) of MED whitefly (Figure 5D and E in Guo et al., 2018). Although mTYLCV can also be detected in the midgut of MED whitefly, it was barely detected in hemolymph (Figure 8D and E in Guo et al., 2018). We rewrote this part: “The mTYLCV, a coat protein mutant of TYLCV in which a partial sequence of CP was replaced by the Papaya leaf curl China virus and hardly penetrated the whitefly gut barrier, was used to determine whether the gut barrier could prevent the impairment of host preference of viruliferous whitefly (Guo et al., 2018; Wei et al., 2017).”

6)The functional genetics analyses combined with (fluorescent) histological assessments look very convincing. We are assuming that the RT-QPCR data does too, though there is not enough information given about the reference genes used, their stability across samples and the efficiency of the primers. This makes it impossible to assess if the RT-QPCR data is valid because these are all things that influence the data presented and if the ΔCT method can be even applied. We are also not sure why the authors log transformed their RT-QPCR data. Is this perhaps a form of p-hacking? If the RT-QPCR data has been dealt with accordingly (which we assume for now) this data is nicely in line with the histology, behavior and functional gene assays.

Sorry for mis-leading you the RT-QPCR part. Actually, we did not transform our QPCR data and the sentence was removed. All CT values were analyzed by relative quantification with the 2^-ΔΔCT^ method, and the replication numbers of the samples in each treatment were annotated in each figure legend. We believed that the ΔCT method was appropriate, because most of our QPCR results were in agreement with our transcriptome data (the excel files of differently expressed genes and all genes FKPM named Supplementary file 2 and Supplementary file 3 were attached in the revised manuscript). We therefore did not further determine the primer efficiency because many of the primers were described in previous documentations, and others were designed by the software Primer Premier 6 with appropriate parameters. The availability of all primers had also been tested in preliminary experiments. (See subsection “PCR and quantitative real-time PCR).

7) The other aspect that is difficult to interpret were the immunodetection protein gels that would confirm caspase activity related to apoptosis. Those gels do not look convincing but it might be partly due to the explanation that the authors provided, which could be better stated.

This comment has been addressed above in point #4, and we tried our best to explain the whole process of effector caspase activation. Please see the details above. We also rewrote the paragraph in our revised manuscript: “Canonical effector caspase cleavage in model species has been well characterized and represents the initiation of apoptosis. […] These results demonstrated that TYLCV virions directly induced caspase3b cleavage in whitefly (Figure 2F).”

8) The authors have not provided enough data and information with regards to their RNASeq experiment. Read size, number of reads, % coverage, number of DEGs and a supplementary file with DEGs does not seem to have been given. The authors should at least report their RNASeq stats. However, they have also not fully explored the dataset it seems, or given the reader more of an interpretation making this just "a list of genes" which is so often said of transcriptomics datasets while a more extensive analysis would make them useful for researchers besides those publishing them. At the moment, the RNASeq data feels more like an add on that doesn't have much to do with the manuscript because the authors mostly go by RT-QPCR finds. The work is pretty complete without the RNASeq data but it seems like this powerful data could be exploited more by doing for instance a WGCNA or PCA + loading plot analysis to get more from it.

We added the details concerning our transcriptome data and rewrote the paragraph: “To determine the difference of gene expression in the nervous system between viruliferous and non-viruliferous whiteflies, the transcriptome of 4 head samples of each treatment (8 in total) were sequenced (RNA-Seq). […] Further qRT-PCR assays confirmed that 10 of 13 these DEGs were down-regulated by TYLCV, indicating that the TYLCV infection had substantial effects on the nervous system of whitefly (Figure 1J).”

A list of DEGs and a list of all genes FPKM values were appended with the revised manuscript. The raw data had been uploaded to NCBI SRA database and the ID is PRJNA606896.

We apologized for not being fully presented the transcriptomic data in previous version, but we thought it was necessary to hold this part in the manuscript, because it pointed out that the sensory defects of viruliferous whitefly may be the consequence of neurodegeneration. Although Bioinformatics was not what we good at, but we could use this dataset to further investigation in our continued work.